# OpenReview forum: "Forewarned is Forearmed:  Harnessing LLMs for Data Synthesis via Failure-induced Exploration"
_ICLR.cc/2025/Conference — ICLR 2025 Poster_

### Official Review · Reviewer_yaDx · 2024-11-04

**Soundness:** 2
**Presentation:** 2
**Contribution:** 3
**Rating:** 6
**Confidence:** 3

**Summary:**

This paper introduces REVERSEGEN, a method for generating training data by identifying model weaknesses through failure-inducing exploration.

**Strengths:**

* Novel approach to data synthesis through failure exploration
* Comprehensive evaluation across three important tasks (safety, honesty, math reasoning)

**Weaknesses:**

* The paper lacks clear justification and motivation for why generating failure-guided data would improve model performance
* No theoretical framework explaining why failure cases would be more valuable than standard training data
* Table 6 shows similar result w and wo failure induction in math reasoning task, does this mean failure induction does not benefit math reasoning tasks?
* No analysis of computational costs or token/API budget comparisons with baseline methods
* Reward mechanism not clearly explained
* Insufficient baseline comparisons, especially for mathematical reasoning task

**Questions:**

* Could you provide theoretical or empirical justification for why failure-inducing data would be more valuable than standard training data for model improvement?
* How does REVERSEGEN's performance evolve with increasing iterations? What determines convergence?
* Could you analyze why failure induction appears less effective for mathematical reasoning (Table 6) compared to other tasks?
* What are the computational requirements (tokens, API calls, training time) compared to baseline methods?
* Could you provide more details about the reward mechanism design and validation process?

---

> ### Author Response · Authors · 2024-11-27
>
> Thank you for your detailed review and for acknowledging our method and comprehensive experiments. We have revised our manuscript to incorporate your valuable suggestions, including:
>
> - An explanation of performance in mathematical reasoning domains in Section 4.4 and Appendix E.1.
> - A clarification of the role of iterative learning in the ReverseGen framework in Figure 2.
> - An analysis of the importance of failure-inducing exploration in Appendix A.
>
> > **W1&W2&Q1: The paper lacks clear justification and motivation for why generating failure-guided data would improve model performance. No theoretical framework explaining why failure cases would be more valuable than standard training data**
>
> For the motivation of failure-inducing exploration, we would like to clarify three sources of evidence from our paper and previous work:
>
> - **Evidence from the target model side**:
>
>     - We have observed that for various tasks such as reasoning and safety tasks, the LLM correctly solves simpler or learned examples in the pre-training dataset but may struggle with harder or not yet learned examples. It will be sub-optimal to keep synthesizing data points for which the LLM is already achieving high accuracy on.
>
>     - Furthermore, different models may excel in various areas or distributions, exhibiting distinct strengths and weaknesses across question types. For instance, we found that LLaMA2-7b-chat often struggles with questions related to privacy, while Vicuna-7b-v1.5 is susceptible to attacks involving harmful tool usage. This variability underscores the importance of generating failure-guided data to address the specific shortcomings of each target model.
>
> - **Evidence from the synthesis model side**: Current synthesized datasets exhibit a severe bias toward easy queries, resulting in low coverage for harder ones [1]. These biases hinder the learning of target models, as challenging examples are often more critical for effective training. This inconsistency results in inferior performance compared to training on human data distributions.
>
> - **Evidence from previous research**: Recent studies have shown that augmenting solutions for difficult samples [1] or providing additional feedback for failure cases [2] can improve model alignment. Our work transforms the task of addressing weaknesses specific to certain models and tasks into a trainable framework.
>
> In response to the reviewer's concerns, we have elaborated the motivation of failure-inducing exploration in Appendix A.
>
> > **W3&Q3: Table 6 shows similar result w and wo failure induction in math reasoning task, does this mean failure induction does not benefit math reasoning tasks? Could you analyze why failure induction appears less effective for mathematical reasoning (Table 6) compared to other tasks?**
>
>
> To investigate the role of challenging samples in target model training, **we maintain a consistent setting across the three tasks**, i.e., employing the same simple supervised fine-tuning to train the target model on the failure-inducing synthesis data for all tasks.
>
> For math task, by only generating 7.9K samples with 1 iteration, the failure-inducing framework can improve Meta-Llama-3-8B-Instruct from 79.89 (8-shot) to 80.93 (0-shot), compared to WizardMath 80.05 (0-shot). This improvement is non-negligible to Meta-Llama-3-8B-Instruct, which is a strong model in mathematics by pretraining over 10M data with upsample math domain data [1].
>
> **We found that the learning efficiency on hard data varies depending on the task**: Simple SFT shows notable enhancements for models in non-deterministic tasks like safety defense or honesty expression. However, in the case of complex deterministic tasks involving multi-step logical reasoning, such as math reasoning, simple SFT struggles to fully leverage the benefits of challenging data without substantial scaling. Based on this analysis, we adopt a difficulty-aware training method [2], the advantages of ReverseGen would become more apparent:
>
> | Setting | GSM8K  | GSMPlus |
> | ----------- | ----------- | ----------- |
> | Initial (8-shot) | 79.89  | 58.78 |
> | WizardMath  | 80.21  | 58.88 |
> | ReverseGen  | 82.26  | 59.92 |
>
> These findings can further verify the effectiveness of ReverseGen in useful challenging data, which is the important basis for the following task performance enhancement.

---

> > ### Author Response · Authors · 2024-11-27
> >
> > > **W4&Q4: No analysis of computational costs or token/API budget comparisons with baseline methods. What are the computational requirements (tokens, API calls, training time) compared to baseline methods?**
> >
> > The core design of ReverseGen is preference training. Our method functions effectively even with just 238 training samples, requiring approximately 2 minutes on 8 Nvidia V100-32GB GPUs. For the safety task, we conduct 3 iterations, resulting in a total training time ranging from 1.3 to 4.27 GPU hours.
> >
> > We use GPT-4o-mini, the most cost-efficient small model provided by OpenAI, to assess instruction quality and label reference responses for knowledge-intensive tasks (question answering and mathematical reasoning). The total cost for this process is approximately $61.44.
> >
> > With limited resources, effectively exploring valuable and challenging data is especially promising for target models that have undergone extensive pre-training and post-training. In this paper, we utilize 8 Nvidia V100-32GB GPUs and GPT-4o-mini to conduct all experiments, successfully generating useful training datasets for three tasks. This contrasts with baselines that rely on human-annotated datasets or multiple prompts to strong LLMs.
> >
> > > **W5&Q5: Reward mechanism not clearly explained. Could you provide more details about the reward mechanism design and validation process?**
> >
> > The reward signals are collected based on “failure” from the target model (Section 3.3). We categorize each instruction generated by the proposer model into the positive set if it induces any errors in the target model; otherwise, it is placed in the negative set.
> >
> > We use a quality indicator $S()$ to detect any errors made by target models. $S()$ represents safety in red-teaming, honesty in question answering, or accuracy in math reasoning tasks. For safety, we use Llama-Guard-2-8B to evaluate whether the generated instructions make target model produce harmful responses. For honesty, we see whether the answer generateg by target model have high entropy values.
> >
> > Specifically, for tasks that require a reference answer, such as mathematical reasoning, we use GPT-4o-mini to label the responses in advance, balancing effectiveness and cost-efficiency. We consider instructions that lead to disagreements between the target model and GPT-4o-mini as effective.
> >
> > In the latest version, we have clarified the validation process in Section 4.1 (highlighted in blue) and summarized the parameters used in each setting in Appendix C. We are happy to provide further clarifications during our discussion.
> >
> >
> > > **W6: Insufficient baseline comparisons, especially for mathematical reasoning task**
> >
> > As suggested by the reviewer, we added an iterative baseline LLM2LLM [2] for comparison, which iteratively prompts proposer from the data points that the target model got incorrect. These results have been incorporated into the main text.
> >
> > | Setting  | GSM8K  | GSMPlus |
> > | ----------- | ----------- | ----------- |
> > | Initial (8-shot)  | 79.89  | 58.78 |
> > |  WizardMath  | 80.05   | 58.75 |
> > | LLM2LLM  | 80.14  | 57.67 |
> > | ReverseGen | 80.93  | 59.31 |
> >
> > We also provide an analysis of the performance of advanced mathematical models in Appendix G.2.
> >
> > > **Q2: How does REVERSEGEN's performance evolve with increasing iterations? What determines convergence?**
> >
> > - Learning from failure samples can be seen as a full exploitation of the weaknesses collected so far (exploitation), and it has been proven effective in aiding dataset synthesis. The iterative optimization of ReverseGen provides an additional benefit by allowing the proposer model to continuously receive new rewards from instructions that can cause target models to fail (exploration).
> >
> > - **Iterative optimization becomes particularly beneficial when initial instructions and failure feedback are limited**. For example, in safety red-teaming, the LLaMA-2-7B-Chat model exhibits significant resilience to attacks. Following the reviewer's suggestion, we conduct four iterations of preference learning to generate harmful instructions for LLaMA-2-7B-Chat. The continuous improvement in ASR highlights the importance of failure induction in data synthesis.
> >
> >     | Iteration | ASR  | Diversity   |
> >     | ----------- | ----------- | ----------- |
> >     | 0  | 0.47  | 0.610  |
> >     | 1  |  0.66  | 0.611  |
> >     | 2  | 2.07  | 0.762 |
> >     | 3  | 8.47  | 0.766 |
> >     | 4  | 27.08  | 0.569 |
> >
> > We revised the manuscript in Section 4.2 to clarify the iteration process.
> >
> > The primary challenge of iteration is finding a policy for effective exploration, which is not trivial [3,4]. In this paper we use the rejection sampling and diversity selection to induce iterative exploration for effective and unique harmful instruction generation. Continuous iterative learning requires a careful balance between exploration and exploitation to prevent potential performance regression [5].

---

> > > ### Author Response · Authors · 2024-11-27
> > >
> > > [1] Yuxuan Tong, Xiwen Zhang, Rui Wang, Ruidong Wu, Junxian He. DART-Math: Difficulty-Aware Rejection Tuning for Mathematical Problem-Solving. https://arxiv.org/abs/2407.13690
> > >
> > > [2] Kai Chen, Chunwei Wang, Kuo Yang, Jianhua Han, Lanqing Hong, Fei Mi, Hang Xu, Zhengying Liu, Wenyong Huang, Zhenguo Li, Dit-Yan Yeung, Lifeng Shang, Xin Jiang, Qun Liu. Gaining wisdom from setbacks: Aligning large language models via mistake analysis. https://arxiv.org/abs/2310.10477
> > >
> > > [3] Nicholas Lee, Thanakul Wattanawong, Sehoon Kim, Karttikeya Mangalam, Sheng Shen, Gopala Anumanchipalli, Michael W. Mahoney, Kurt Keutzer, Amir Gholami. LLM2LLM: Boosting LLMs with Novel Iterative Data Enhancement. https://arxiv.org/abs/2403.15042
> > >
> > > [4] Hanze Dong, Wei Xiong, Bo Pang, Haoxiang Wang, Han Zhao, Yingbo Zhou, Nan Jiang, Doyen Sahoo, Caiming Xiong, Tong Zhang. RLHF Workflow: From Reward Modeling to Online RLHF. https://arxiv.org/abs/2405.07863
> > >
> > > [5] Llama Team, AI @ Meta. The Llama 3 Herd of Models. https://arxiv.org/abs/2407.21783

---

> > > > ### Author Response · Authors · 2024-11-30
> > > >
> > > > Dear Reviewer yaDx,
> > > >
> > > > We would appreciate it if you could let us know whether our response has adequately addressed your concerns and questions. We remain available to address any further questions you may have.
> > > >
> > > > Thank you once again for your time and effort!

---

> > > > > ### Author Response · Authors · 2024-12-01
> > > > >
> > > > > Dear Reviewer yaDx,
> > > > >
> > > > > Thank you for your thoughtful feedback and for acknowledging the merit of our method! We have revised our manuscript and clarified your questions in response to your comments.
> > > > >
> > > > > If our responses address your initial concerns, we would greatly appreciate it if you could consider updating your assessment.
> > > > > Thank you again for your time.

---

> > > > > > ### Author Response · Authors · 2024-12-02
> > > > > >
> > > > > > Dear Reviewer yaDx,
> > > > > >
> > > > > > As the discussion period approaches its closing date, we wanted to follow up to see if our responses have addressed your concerns. We have incorporated your suggested experimental analyses and are open to continuing the conversation if needed.
> > > > > >
> > > > > > We kindly hope you will reconsider your initial assessment. Thank you in advance for your recognition!

---

> ### Author Response · Authors · 2024-12-02
>
> Dear Reviewer yaDx,
>
> We sincerely hope you can reconsider **your initial assessment (3: reject, not good enough)**, especially with less than 23 hours remaining until the rebuttal deadline.
>
> In response to your constructive feedback, we have addressed your questions, provided detailed experimental analyses, and revised our manuscript, including:
>
> - An explanation of performance in mathematical reasoning domains in Section 4.4 and Appendix G.1.
> - A clarification of the role of iterative learning in the ReverseGen framework in Figure 2 and Appendix E.2.
> - An analysis of the importance of failure-inducing exploration in Appendix A.
> - A clear experimental configuration including reward designs in Section 4.1 and Appendix C.
>
> We are open to further discussion if needed. Thank you for your support!
>
> Best regards,
>
> The Authors of 8852

---

> > ### Comment · Reviewer_yaDx · 2024-12-02
> >
> > Thanks for the reply. I'm raising my score to 6.

---

> > > ### Author Response · Authors · 2024-12-03
> > >
> > > Dear Reviewer yaDx,
> > >
> > > Thank you again for your valuable feedback, which helped us refine the quality of our paper!

---

### Official Review · Reviewer_HFrR · 2024-11-04

**Soundness:** 2
**Presentation:** 3
**Contribution:** 3
**Rating:** 6
**Confidence:** 4

**Summary:**

The paper proposes a technique called ReverseGen for generating failure-inducing examples for an LLM. The technique uses a proposer model that is fine-tuned using pairs of positive and negative examples. The evaluation shows that the generated data can be used to fine-tune and improve models in safety, honesty, and mathematical reasoning.

**Strengths:**

- The paper introduces a new technique for synthesizing failure-inducing data for a target LLM

- The technique is effective in 3 distinct domains of safety, honesty, and mathematical reasoning and shows improvement on SOTA on each. I appreciate the inclusion of Table 5 and Table 7 with examples in the evaluation section

**Weaknesses:**

The writing of the paper, especially the separation of the technical section and implementation details needs to be improved. Many of the evaluation choices seem a bit arbitrary and need to be organized better to understand how each of the applications of the proposed technique fits into a single framework (if they do).

- Line 126: “ these studies tend to focus on particular tasks or inputs, and overlook the task generalization and the data salience on model enhancement. Our work differs from these prior works in that we focus on generating failure-guided data for target models”
Can you be more specific in comparison with each of these prior works? It is unclear to me how ReverseGen differs from many of these works mentioned in the related works


- Line 176: The term “solvable” and “unsolvable” is defined here and never mentioned here. The term “solvable” for positive examples is quite unclear. Use a more appropriate name for this.


- Line 202: Section 3.2 is the technical section and it starts talking about “gpt-40-mini”. I would recommend authors to separate the implementation details from the technical section. “Gpt-40-mini” is a specific model and used to label the responses, define the model used for labeling as a parameter of the technique that’s instantiated as a specific model.

## Minor:
- Line 189: “We begin by warming up the proposer model $M_{prop}$ with the initial task-specific instruction set” - A bit unclear wording. Can be more technically precise, especially in the technical section of the paper

- Line 190: “3-shot prompts” - this seems like implementation detail as well which would be more appropriate if it was in the evaluation section

- Line 204: What’s R(.)? I don’t see it defined anywhere before

- Line 220: “$M_{ref}$ is the reference model that remains unchanged”: this doesn’t really define what is $M_{ref}$

- Line 227: typo - two dots “..”

- Line 275: why were these hyperparameters chosen?

- Line 284: “Responses from the target model are generated using greedy decoding for deterministic quality measurement, with the temperature parameter set to 0” - Doesn’t greedy already mean temperature doesn’t matter?

- Line 287: “Instructions for the SFT data are produced by proposer models, while responses are derived from the target models using tailored prompts for safety tasks, generated by gpt-4o-mini for knowledge-intensive tasks.” -> the sentence is too long and hard to understand. What are “knowledge-intensive tasks” in this context?

- Line 319: “well-safe”?

- Line 429: “ReverseGen solely relies on synthetic data” - Doesn’t it use MMLU as the initial instruction seed?

- Line 523: “This may result from inadequate assessment of REVERSEGEN’s proposed questions by the current benchmark or the need for more effective fine-tuning algorithms for difficult questions.” - a bit unclear

**Questions:**

Since the technical section and implementation details are somehow mixed, it is really hard to understand some of the details of each of the experiments. I have the following questions about the evaluation:

- In what experiments, GPT-40-mini used? How was this decided? Is there an ablation study on its use?
- For the honesty calibration and mathematical reasoning experiment, what number of "proposer fine-tuning" iterations are used in ReverseGen?
- Why is Llama3 used in 4.4 but not in the previous experiments?

Can you add a table that maps each of the variables in the technical section to the specific choice made in each experiment? This would make it easier for the reader to understand each experiment. For instance, the columns could be $M_{prop}, M_{tgt}$, number of examples, seed data, number of iterations, use of GPT-4o-mini, and other implementation details for each experiment.

---

> ### Author Response · Authors · 2024-11-27
>
> Thank you for your feedback and constructive comments. And thanks for your recognition of the novelty and effectiveness of our method.
>
> We have revised our manuscript according to your suggestions, including:
>
> - Rephrasing and reorganizing the confusing sentences in the technical section (Section 3) and move the pseudocode to Appendix B. The main revisions are highlighted in blue.
> - Incorporation of a table to clearly map each of the variables in the technical section to the specific choice made for each task in Appendix C.
>
> _________________
>  We address your concerns below.
>
> > **W1.1 Line 126: “... Our work differs from these prior works in that we focus on generating failure-guided data for target models” Can you be more specific in comparison with each of these prior works? It is unclear to me how ReverseGen differs from many of these works mentioned in the related works.**
>
> Thank you for your suggestion! We have clarified and expanded these statements for better understanding in Section 2.1.
>
> Most prior research on generating data to address the weaknesses of target models has focused on known instructions, specific tasks, or distilling knowledge from strong LLMs using prompts.
>
> Our work differs from prior research by transforming the feedback process into a trainable framework focused on synthesizing failure-inducing data. We optimize the model by rewarding it for generating instructions that lead to failures in the target model. The incorporation of an iterative preference learning algorithm enables the model to continuously learn from newly generated instructions that challenge the target model.
>
> We hope the current version clarifies the distinctions more clearly.
>
> > **W1.2 Line 176: The term “solvable” for positive examples is quite unclear. Use a more appropriate name for this.**
>
> Thank you for raising this point.  We have changed "solvable" to "ineffective", which means these instructions do not cause the target model to fail. We also changed "unsolvable" to "effective", referring to instructions that successfully challenge the target model.
>
> Minor
>
> > **Line 275: why were these hyperparameters chosen?**
>
> We use the default hyperparameters from HALOs (https://github.com/ContextualAI/HALOs), a library that implements preference learning algorithms. We modified the batch size to fit our hardware requirements.
>
>  > **Line 429: “ReverseGen solely relies on synthetic data” - Doesn’t it use MMLU as the initial instruction seed?**
>
> Yes, we optimize the target model solely with data generated by ReverseGen, achieving superior performance compared to the target model trained on human data.
>
> > **Line 284: “Responses from the target model are generated using greedy decoding for deterministic quality measurement, with the temperature parameter set to 0” - Doesn’t greedy already mean temperature doesn’t matter?**
>
> To accelerate data generation, we use VLLM (https://docs.vllm.ai/en/stable/dev/sampling_params.html) for inference. In VLLM, temperature parameter set to zero means greedy sampling.

---

> > ### Author Response · Authors · 2024-11-27
> >
> > > **Q1: In what experiments, GPT-40-mini used? How was this decided? Is there an ablation study on its use?**
> >
> > For knowledge-intensive tasks such as honesty calibration and mathematical reasoning, we use GPT-4o-mini to (1) evaluate the correctness of the instructions generated by the proposer model and (2) label the correct answers for qualified instructions. This process is essential for creating effective samples that address the weaknesses of the target model.
> >
> > We chose GPT-4o-mini because it is the most cost-efficient small model [1], widely adopted in LLM-as-a-judge and data labeling scenarios. We believe GPT-4o-mini is reliable for both instruction evaluation and instruction-following tasks.
> >
> > We can also use the strong open-source model. For example, Qwen2.5-Math-7B is a substitute for OpenAI models in mathematical tasks. Qwen2.5-Math-7B achieves an 8-shot score of 91.6 on GSM8K, comparable to GPT-4o's score of 92.9 on the same benchmark. We found that Qwen2.5-Math-7B provides comparable judgments to GPT-4o-mini, with approximately 86.4% agreement across 7.9K samples.
> >
> > > **Q2: For the honesty calibration and mathematical reasoning experiment, what number of "proposer fine-tuning" iterations are used in ReverseGen?**
> >
> > We use one iteration of preference learning from target model feedback.
> >
> > > **Q3: Why is Llama3 used in 4.4 but not in the previous experiments?**
> >
> > We chose Llama-3-8B-Instruct because it performs exceptionally well on standard math benchmarks compared to models used for safety red-teaming and honesty calibration. However, it can make simple errors and often struggles with specific math questions, such as deriving variable relationships. This creates a new challenge for data synthesis methods to expose its weaknesses and generate training samples for improvement. We show that ReverseGen is effective in this challenging context.
> >
> >
> > [1] https://openai.com/index/gpt-4o-mini-advancing-cost-efficient-intelligence/

---

> > > ### Author Response · Authors · 2024-11-30
> > >
> > > Dear Reviewer HFrR,
> > >
> > > We would appreciate it if you could let us know whether our response has adequately addressed your concerns and questions. We remain available to address any further questions you may have.
> > >
> > > Thank you once again for your time and effort!

---

> > > > ### Comment · Reviewer_HFrR · 2024-11-30
> > > >
> > > > > Q2: For the honesty calibration and mathematical reasoning experiment, what number of "proposer fine-tuning" iterations are used in ReverseGen? A. We use one iteration of preference learning from target model feedback.
> > > >
> > > > The paper emphasizes on an iterative refinement framework. However, 2 out of 3 evaluation experiments only use single iteration of refinement. Am I missing something here?

---

> > > > > ### Author Response · Authors · 2024-11-30
> > > > >
> > > > > We sincerely appreciate your feedback regarding the iterative optimization process.
> > > > >
> > > > > **In the context of safety red-teaming, initial failure feedback is often limited, making iterative preference learning especially valuable.** For instance, while generating harmful instructions for Llama-2-7b-chat using OpenLLaMA-3B, we only collected 238 effective training samples from 50K prompts, resulting in a 1.40x increase in ASR at t=1. In comparison, at t=2, we gathered 380 samples, achieving a 3.56x increase in ASR, and at t=3, with 598 samples, we realized a 4.44x increase in ASR.
> > > > >
> > > > > **For honesty calibration and mathematical reasoning, collecting failure cases is much easier.** After initialization, we obtained 1,504 effective instructions that caused uncalibrated responses from 50K prompts. By training on these instructions with one iteration, 8.77% of the generated instructions were labeled as “effective,” leading to target model failures. This efficiency allows us to scale the training samples and significantly improve the calibration score of the target models.
> > > > >
> > > > > To address the review’s follow-up question, we pursue further iterative optimization. At t=2 and t=3, the updated proposer model produced 10.98% and 14.88% effective instructions, respectively. Through qualitative analysis, we observed that the updated proposer model tends to generate questions related to high school chemistry and abstract algebra, exposing uncalibrated behaviors of the target model in those areas. This finding underscores the promising potential of failure feedback in data synthesis. We will add more analyses to the later version of this paper.
> > > > >
> > > > > We hope this addresses your follow-up question. Thank you again for your time.

---

> > > > > > ### Comment · Reviewer_HFrR · 2024-11-30
> > > > > >
> > > > > > Thank you for the response. The changes made in the rebuttal have improved the paper's readability, and I am raising my presentation score from 2 to 3.
> > > > > >
> > > > > > However, my overall score remains contingent on receiving satisfactory answers to some questions that are still unresolved. It would be helpful if the authors were more explicit in addressing some of the comments.
> > > > > >
> > > > > > > Line 429: “ReverseGen solely relies on synthetic data” - Doesn’t ReverseGen use MMLU as the initial instruction seed?
> > > > > >
> > > > > > The rebuttal response does not answer my question. Doesn’t ReverseGen use MMLU as the initial instruction seed and not depend solely on synthetic data?
> > > > > >
> > > > > > > Q3: Why is Llama3 used in 4.4 but not in the previous experiments?
> > > > > >
> > > > > > I realize I should have been more explicit in phrasing this question. My main inquiry is: how were the models selected for each experiment? For instance, the "Safety Red-teaming" and "Honesty Calibration" experiments utilized older versions of the Llama models, while the "Mathematical Reasoning" experiment employed the newer Llama-3 model. Why didn’t the authors use the Llama-3 models consistently across all experiments?
> > > > > > Additionally, how were the specific proposer models chosen for each task?
> > > > > > My primary concern is ensuring that the technique demonstrates robustness across a variety of model choices and does not rely on cherry-picked model and proposer combinations that happen to produce favorable results.
> > > > > >
> > > > > > > Line 284: “Responses from the target model are generated using greedy decoding for deterministic quality measurement, with the temperature parameter set to 0” - Doesn’t greedy already mean temperature doesn’t matter?
> > > > > >
> > > > > > This comment indicated that one need not say the temperature is set to 0 when greedy decoding is used. Regardless of the generation framework used, the temperature hyperparameter is meaningless in greedy decoding. Authors should fix this line in the next revision of the paper and remove the part that says the temperature is set to 0.
> > > > > >
> > > > > > Other minor nits:
> > > > > >
> > > > > > Line 128: Fix citation style
> > > > > >
> > > > > > Line 381: Emplots -> Employs
> > > > > >
> > > > > > Line 927: “Safety Responses, generated by the target model itself, gpt-4o-mini, through promptings” - I cannot understand this sentence

---

> ### Author Response · Authors · 2024-12-01
>
> > **Line 429: “ReverseGen solely relies on synthetic data” - Doesn’t ReverseGen use MMLU as the initial instruction seed? The rebuttal response does not answer my question. Doesn’t ReverseGen use MMLU as the initial instruction seed and not depend solely on synthetic data?**
>
> Thank you for your question, and we apologize for any misunderstanding.
>
> During the initialization phase, we use multiple-choice questions from the MMLU training set as the initial instruction seed, which is essential for adapting the proposer model to the MMLU distribution.
>
> When enhancing the target model, we do not use any MMLU training data. Instead, we rely solely on samples generated by the proposer model for performance calibration, which yields better results than the baseline that uses the MMLU training set.
>
> > **Q3: Why is Llama3 used in 4.4 but not in the previous experiments? I realize I should have been more explicit in phrasing this question. My main inquiry is: how were the models selected for each experiment? For instance, the "Safety Red-teaming" and "Honesty Calibration" experiments utilized older versions of the Llama models, while the "Mathematical Reasoning" experiment employed the newer Llama-3 model. Why didn’t the authors use the Llama-3 models consistently across all experiments? Additionally, how were the specific proposer models chosen for each task? My primary concern is ensuring that the technique demonstrates robustness across a variety of model choices and does not rely on cherry-picked model and proposer combinations that happen to produce favorable results.**
>
> **Selection of Target Models**: In safety red-teaming, we focused on Llama-2-7b-chat due to its extensive pre-training and post-training efforts related to safety, which enable it to demonstrate strong safety defense performance, even when compared to the later Llama-3-8B-Instruct [1]. To maintain consistency with the baseline Curiosity [2], we selected Vicuna-7b-v1.5 and Llama-2-7b-chat as our target models. We focused on Llama-3 within the math domain due to its significant improvements in reasoning tasks compared to the previous Llama 2 models [3].
>
> - To address the reviewer’s concern, we perform a further evaluation of the harmful instructions generated by OpenLLaMA-3B for attacking Llama-3-8B-Instruct. Similar to ReverseGen with Llama-2-7b-chat, we collected 520 effective harmful instructions that successfully targeted Llama-3-8B-Instruct. With one iteration, we improved the ASR from 1.03% to 2.45%, and with two iterations, we enhanced the ASR to 7.34%. These results demonstrate the generalization and robustness of ReverseGen across different target models.
>
> **Selection of Proposer Models**: We aim to investigate whether we can depend on tunable open-source models to generate effective instructions, rather than distilling instructions from stronger models or human. To achieve this, there are two key requirements for the proposer models:
> 1. they must be capable of generating task-related instructions based on few-shot prompts;
> 2. they should be pretrained on domain-related data to ensure the quality of the generated instructions.
>
> With these criteria in mind, we can flexibly combine proposer models with target models. We will include these analyses in the revised version of the paper.
>
> > **This comment indicated that one need not say the temperature is set to 0 when greedy decoding is used. Regardless of the generation framework used, the temperature hyperparameter is meaningless in greedy decoding. Authors should fix this line in the next revision of the paper and remove the part that says the temperature is set to 0.**
>
> We appreciate the reviewer’s suggestion and will remove this statement in the next revision to improve accuracy.

---

> ### Author Response · Authors · 2024-12-01
>
> > **Line 128: Fix citation style**
>
> Thanks for your comment. We have revised the citation style.
>
> > **Line 381: Emplots -> Employs**
>
> Thank you for the reminder. We have corrected the typo.
>
> > **Line 927: “Safety Responses, generated by the target model itself, gpt-4o-mini, through promptings” - I cannot understand this sentence**
>
> To prepare the training data for enhancing the safety capabilities of target models, we use crafted prompts to induce them to generate safety responses for harmful instructions generated by ReverseGen (lines 1009-1010). Although we could use GPT-4o-mini to produce these responses, our prior experiments show that self-generated responses provide slightly better defense performance. We will clarify this in the revised version.
>
> _________________
> We really appreciate receiving these valuable and constructive comments from the reviewer HFrR.
>
> We hope the responses above address your follow-up questions. Thank you again for your time and feedback.
>
> [1] Maksym Andriushchenko, Francesco Croce, Nicolas Flammarion. Jailbreaking Leading Safety-Aligned LLMs with Simple Adaptive Attacks. https://arxiv.org/abs/2404.02151
>
> [2] Zhang-Wei Hong, Idan Shenfeld, Tsun-Hsuan Wang, Yung-Sung Chuang, Aldo Pareja, James Glass, Akash Srivastava, Pulkit Agrawal. Curiosity-driven Red-teaming for Large Language Models. https://arxiv.org/abs/2402.19464
>
> [3] https://ai.meta.com/blog/meta-llama-3/

---

> > ### Author Response · Authors · 2024-12-02
> >
> > Dear Reviewer HFrR,
> >
> > We greatly appreciate your follow-up feedback, which has been very helpful for our paper.
> >
> > As the discussion period approaches its closing date, we wanted to check in to see if our responses have addressed your questions. We’re happy to continue the conversation if needed.
> >
> > Thank you in advance for your support!

---

> ### Comment · Reviewer_HFrR · 2024-12-02
>
> Thanks for the reply. I'm raising my score to 6.
>
> I had one more question about the additional experiment:
>
> > we perform a further evaluation of the harmful instructions generated by OpenLLaMA-3B for attacking Llama-3-8B-Instruct
>
> Why was OpenLLaMA-3B used in this additional experiment?
> I'd say the experiments will be much more convincing if the authors show an ablation with 7-8 different combinations of target and proposer model in the next revision.

---

> ### Author Response · Authors · 2024-12-02
>
> Dear Reviewer HFrR,
>
> Thank you for your valuable comment.
>
> We chose OpenLLaMA-3B as the proposer model for attacking Llama-3-8B-Instruct due to its comparatively weaker general capabilities relative to other proposer models considered in this paper (e.g., Llama-2-7b). Due to time constraints, we investigated whether OpenLLaMA-3B in ReverseGen can identify increasingly effective instructions for the stronger Llama-3-8B-Instruct through iterations. Our goal is to gain insights into the flexibility and generalization of our framework.
>
> We will explore additional combinations of proposer and target models as soon as possible and include those experimental results in the next revision. We greatly appreciate your understanding.
>
> Once again, we would like to express our sincere gratitude to you for these constructive suggestions.
>
> Best regards,
>
> The Authors of Paper 8852

---

### Official Review · Reviewer_29jz · 2024-11-04

**Soundness:** 2
**Presentation:** 3
**Contribution:** 2
**Rating:** 6
**Confidence:** 4

**Summary:**

This paper proposes a data synthesis approach by training a data generator and leverage the performance of the target model as a training signal. Specifically, the predictions of the target model are used to construct a preference dataset (target model's failure cases are preferred) for the training of the data generator which performs DPO on top of those preference data.

**Strengths:**

* The paper is overall well written and easy to understand.
* The proposed approach is novel. It performs RLHF to train the data generator, with the target model works as a preference provider.

**Weaknesses:**

The experiments miss 2 significant baselines.
* To verify the effect of the proposed RLHF approach, there should a baseline finetuning the data generator (proposer LLM) with a collection of failed samples, and generate a dataset.
* A strong LLM (i.e., gpt-4o) plays an important role in the proposed method when obtaining the oracle label, so there should another baseline directly prompting the gpt-4o multiple times to generate a synthetic dataset.

**Questions:**

None

---

> ### Author Response · Authors · 2024-11-27
>
> Thank you for your thoughtful review and comment. And thanks for your recognition of the writing and novelty of our method.
> In response to your suggestion, we have added an analysis on directly distilling from powerful mathematical models in Appendix G.2.
>
> _________________
> We address the concerns below:
>
> > **W1: To verify the effect of the proposed RLHF approach, there should a baseline finetuning the data generator (proposer LLM) with a collection of failed samples, and generate a dataset.**
>
> We do not assume the target model will fail with the initial instruction set. Even when starting with a mixed or unbalanced instruction set to warm up the proposer model, ReverseGen can progressively generate training samples that cause target model failures.
>
> As suggested by the reviewer, we compare two initialization settings before conducting preference optimization: (1) **Mix**: Training with randomly sampled instructions (a combination of easy and difficult data); (2) **Difficulty**: Training with randomly sampled hard instructions for the target model.
>
>
> - For safety red-teaming (target: Vicuna-7b-v1.5):
>
>     | Proposer | ASR| Diversity|
>     | ----------- | ----------- | ----------- |
>     | OpenLLaMA-3B (Mix) | 6.68 |  0.571 |
>     | OpenLLaMA-3B (Difficulty) | 12.62 | 0.313 |
>     | ReveseGen (t=1) | 19.76 | 0.596 |
>     | ReveseGen (t=2) | 32.83  | 0.620 |
>     | ReverseGen (t=3) | 56.73 | 0.465 |
>
>     In safety red-teaming, preparing sufficient harmful instructions in advance is difficult. Therefore, we prepare the training data for the Difficulty setting by using the same in-context demonstrations as in the Mix setting to induce the proposer model to generate instructions and collect harmful instructions for warm-up.
>
> - For mathematical reasoning:
>
>     | Proposer | GSM8K  | GSMPlus |
>     | ----------- | ----------- | ----------- |
>     | Llama-3-8B (Mix)  | 77.96  | 58.29 |
>     | Llama-3-8B (Difficulty)  | 81.44   | 59.00 |
>     | ReverseGen  | 80.93  | 59.31 |
>
>     Training begins with a high-quality set of challenging instructions, which provides a strong initial policy for further preference learning. We use this as the initialized policy for ReverseGen when sufficient training data is available (e.g., in the math domain). However, currently the main bottleneck for math optimization lies in developing an efficient difficulty-aware algorithm, which we analyze in detail in Appendix G.1.
>
> **Summary**:
>
> 1. Initializing with challenging samples can help us establish a strong initial policy if we have sufficient failure cases for training.
>
> 2. However, collecting numerous failure cases is often impractical in many real-world domains. Despite this, ReverseGen remains effective even when the initial seeds are not necessarily challenging instructions.
>
>
> > **W2: A strong LLM (i.e., gpt-4o) plays an important role in the proposed method when obtaining the oracle label, so there should another baseline directly prompting the gpt-4o multiple times to generate a synthetic dataset.**
>
> Owing to restricted API resources, we have chosen the robust open-source model Qwen2.5-Math-7B for the reviewer's reference. Qwen2.5-Math-7B achieves an 8-shot score of 91.6 on GSM8K, a score comparable to GPT-4o's performance of 92.9 on the identical task.
>
> | Proposer | GSM8K  | GSMPlus |
> | ----------- | ----------- | ----------- |
> | Qwen2.5-Math-7B (Mix)  | 78.39  | 54.12  |
> |  Qwen2.5-Math-7B (Difficulty)  | 79.37  | 55.00  |
> | ReverseGen  | 80.93  | 59.31 |
>
> **Mix**: Prompting with randomly sampled instructions.
>
> **Difficulty**: Prompting with randomly sampled hard instructions for the target model.
>
> Relying on prompting is insufficient for effectively exploring challenging instructions without preference learning from target model feedback, resulting in subpar performance on the standard benchmark GSM8K and the adversarial benchmark GSMPlus. We will include prompting for GPT-4o-mini in later versions.

---

> > ### Author Response · Authors · 2024-11-30
> >
> > Dear Reviewer 29jz,
> >
> > We would appreciate it if you could let us know whether our response has adequately addressed your concerns and questions. We remain available to address any further questions you may have.
> >
> > Thank you once again for your time and effort!

---

> > > ### Author Response · Authors · 2024-12-02
> > >
> > > Dear Reviewer 29jz,
> > >
> > > As the discussion period approaches its closing date, we wanted to follow up to see if our responses have addressed your concerns. We have added the baseline analyses you suggested and are happy to continue the conversation if needed.
> > >
> > > Thank you in advance for your support!

---

> > ### Author Response · Authors · 2024-12-03
> > **Update on Results for Prompting GPT-4o-mini**
> >
> > Dear Reviewer 29jz,
> >
> > We have completed experiments by prompting GPT-4o-mini to generate challenging questions. Below is an example of a prompt example, which is the same to that used for Qwen2.5-Math-7B. We add an additional system message to avoid verbose expressions.
> >
> > ```json
> > Follow the demonstrations and ask the fourth question. Do not generate any verbose information.
> >
> > Ask math questions:
> > Q1. A drink vendor has 80 liters of Mazza, 144 liters of Pepsi, and 368 liters of Sprite. He wants to pack them in cans, so that each can contains the same number of liters of a drink, and doesn't want to mix any two drinks in a can. What is the least number of cans required?
> >
> > Q2. A sum fetched a total simple interest of Rs. 4016.25 at the rate of 5% p.a. in 5 years. What is the sum?
> >
> > Q3. There is food for 760 men for 22 days. How many more men should join after two days so that the same food may last for 19 days more?
> >
> > Q4.
> > ```
> > | Proposer | GSM8K  | GSMPlus |
> > | ----------- | ----------- | ----------- |
> > |Qwen2.5-Math-7B (Mix)   | 78.39   | 54.12 |
> > | Qwen2.5-Math-7B (Difficulty)  | 79.37  | 55.00  |
> > | GPT-4o-mini (Difficulty) | 75.82 | 55.63 |
> > | ReverseGen | 80.93 | 59.31 |
> >
> > We found that using prompting for data synthesis to enhance Llama-3-8B-Instruct is less effective than ReverseGen, which learns failure signals directly and iteratively. Furthermore, the questions generated by prompting GPT-4o-mini are significantly shorter than the in-context prompt samples (19.05 vs. 29.23).
> >
> > **Summary**: Relying on prompting is insufficient for effectively exploring challenging instructions without preference learning from target model feedback, resulting in subpar performance on the standard benchmark GSM8K and the adversarial benchmark GSMPlus.

---

> ### Author Response · Authors · 2024-12-03
>
> Dear Reviewer 29jz,
>
> Thank you for your constructive comment and for acknowledging our method's clear presentation and novelty. We sincerely hope you can reconsider updating your assessment, especially with **less than 12 hours** remaining until the rebuttal deadline.
>
> We have revised our manuscript and clarified your questions in response to your comments, including
>
> - adding an analysis on directly distilling from powerful mathematical models in Appendix G.2.
> - comparing the difference by initializing on difficulty-prioritized instructions or random task instructions.
> - explaining the performance of prompting GPT-4o-mini or Qwen2.5-Math-7B for data generation.
>
> We are open to further discussion if needed. Thank you in advance for your recognition!
>
> Best regards,
>
> The Authors of 8852

---

### Official Review · Reviewer_u19j · 2024-11-04

**Soundness:** 3
**Presentation:** 3
**Contribution:** 3
**Rating:** 6
**Confidence:** 4

**Summary:**

The authors propose an iterative finetuning method for finetuning a target langague model by using synthetic data generation from a proposer language model which proposes harder and harder questions to a target language model. This is in effect a curriculum learning approach which trains a target model on harder and harder samples. The proposer model is also trained to propose harder and harder questions by using errors in the target model’s answers.

A proposer language model generates few-shot candidate questions. Then the target model predicts answers to these questions. The answers are then compared to gpt-4o-mini's answers, which is used as a gold-standard. If the answers agree then this question is placed into the negative set {x^-}. If the answers from the target model does not agree with the gpt-4o-mini then the question is placed in the positive set {x^+}. These sets are then used by DPO is used to finetune the proposer model to produce harder and harder samples by leveraging the positive and negative sets. Finally, the proposer model generates synthetic questions which are deemed hard for the target model, labels are generated by the proposer model or gpt-4o-mini. The target model is then trained with SFT.

**Strengths:**

- The idea of using the language model prediction errors to create a set of easy and hard examples to train a proposer model with DPO is novel and a nice idea.

- Interesting results that show that harder to predict data points aka using a curriculum harder and harder questions is beneficial for some red teaming and honesty benchmarks. This is interesting since, in comparison other papers such as [1], show that hard samples actually hurts target model performance albeit in a different dataset domain.

- Wide range of experiments: red teaming, honesty and mathematical reasoning to demonstrate that the method can generalize to multiple domains.

[1] Evans, Talfan, et al. "Bad students make great teachers: Active learning accelerates large-scale visual understanding." arXiv preprint arXiv:2312.05328 (2023).

**Weaknesses:**

- Weak results on mathematical reasoning which do not demonstrate considerable improvement in performance. Nor are many other similar iterative methods which have some sort of synthetic question-answer prioritization compared to [2, 3].

- No ablation experiments, what is the performance with 1 iteration versus 5? What if you generate 1k or 10k samples to populate the positive and negative sets for DPO training?

[2] Lee, Nicholas, et al. "Llm2llm: Boosting llms with novel iterative data enhancement." arXiv preprint arXiv:2403.15042 (2024).
[3] Jiang, Yuxin, et al. "Lion: Adversarial distillation of proprietary large language models." arXiv preprint arXiv:2305.12870 (2023).

**Questions:**

- What is $R(\cdot)$ in line 204?

- Performance for ReverseGen in Tables 1 and 2 has not saturated with increasing iterations and it seems like you are under reporting results. What would happen if you ran this for 4 or 5 iterations? At what point would performance saturate? What if you generated 10000k instruction candidates and performed t=1 iteration versus 2000k instruction candidates with t=5 iterations?

- When using harder and harder samples to train a model for example in active learning, or data selection, where data points are prioritized with a larger loss then this can cause a negative feedback loop with a catastrophic drop in performance from a target model. Did you observe similar artifacts in your experiments?

- In Tables 1, 2, 3 you have a rows ‘without failure induction’ but you do not describe what this ablation is?

---

> ### Author Response · Authors · 2024-11-27
>
> Thank you for your careful review as well as appreciation of our work.
>
> According to your suggestions, we add:
>
> - An explanation of performance in mathematical reasoning domains in Section 4.4 and Appendix E.1.
> - A clarification of the role of iterative learning in the ReverseGen framework in Figure 2.
>
> _________________
>
> We address the concerns below:
>
> > **W1: Limited improvements in mathematical reasoning. Nor are many other similar iterative methods which have some sort of synthetic question-answer prioritization.**
>
> To investigate the role of challenging samples in target model training, **we maintain a consistent setting across the three tasks**, i.e., employing the same simple supervised fine-tuning to train the target model on the failure-inducing synthesis data for all tasks.
>
> For math task, by only generating 7.9K samples with 1 iteration, the failure-inducing framework can improve Meta-Llama-3-8B-Instruct from 79.89 (8-shot) to 80.93 (0-shot), compared to WizardMath 80.05 (0-shot). This improvement is non-negligible to Meta-Llama-3-8B-Instruct, which is a strong model in mathematics by pretraining over 10M data with upsample math domain data [1].
>
> **We found that the learning efficiency on hard data varies depending on the task**: Simple SFT shows notable enhancements for models in non-deterministic tasks like safety defense or honesty expression. However, in the case of complex deterministic tasks involving multi-step logical reasoning, such as math reasoning, simple SFT struggles to fully leverage the benefits of challenging data without substantial scaling.
>
> If we apply a difficulty-aware training method [2], the advantages of ReverseGen would become more apparent:
>
> | Setting | GSM8K  | GSMPlus |
> | ----------- | ----------- | ----------- |
> Initial (8-shot) | 79.89  | 58.78 |
> | WizardMath  | 80.21  | 58.88 |
> | LLM2LLM | 80.97  | 56.54 |
> | ReverseGen  | 82.26  | 59.92 |
>
> As suggested by the reviewer, we add an iterative baseline LLM2LLM for comparison, which solely uses hard seed examples as in-context example for generating similar data points. WizardMath generates samples along the knowledge-difficulty level without considering target model. In comparison, ReverseGen balances exploitation and exploration by utilizing failure feedback as a reward to guide the exploration of a learnable generator. We have incorporated these analyses into Section 4.4 and Appendix F.1.

---

> > ### Author Response · Authors · 2024-11-27
> >
> > > **W2&Q2: Ablation experiments**
> >
> >
> > >> **Iteration times. What is the performance with 1 iteration versus 5? Iteration times. What is the performance with 1 iteration versus 5? Performance for ReverseGen in Tables 1 and 2 has not saturated with increasing iterations. At what point would performance saturate?**
> >
> > ReverseGen, even with only one iteration, can effectively explore instructions that highlight the weaknesses of target models, aiding in their enhancement (as shown in Table 3 and Table 5). We want to address reviewer’s concern from three perspectives:
> >
> > - **Iterative optimization is orthogonal to failure-inducing optimization.** Learning from failure samples can be seen as a full exploitation of the weaknesses collected so far (exploitation), and it has been proven effective in aiding dataset synthesis. The iterative optimization of ReverseGen provides an additional benefit by allowing the proposer model to continuously receive new rewards from instructions that can cause target models to fail (exploration).
> >
> > - **Iterative optimization becomes particularly beneficial when initial instructions and failure feedback are limited.** For example, in safety red-teaming, the LLaMA-2-7B-Chat model, which shows significant resilience to attacks. With three iterations, we observed a significant increase in Attack Success Rate (ASR), showing improvements of 1.40x, 3.56x, and 4.44x over previous iterations, all while maintaining diversity. As suggested by the reviewer, we conduct further iteration on generating harmful instructions for Llama-2-7b-chat in safety red-teaming. The continuous improvement in ASR demonstrates the importance of failure-inducing in data synthesis.
> >
> >     | Iteration | ASR  | Diversity   |
> >     | ----------- | ----------- | ----------- |
> >     | 0  | 0.47  | 0.610  |
> >     | 1  |  0.66  | 0.611  |
> >     | 2  | 2.07  | 0.762 |
> >     | 3  | 8.47  | 0.766 |
> >     | 4  | 27.08  | 0.569 |
> >
> >     We revised the manuscript in Section 4.2 to clarify the iteration process.
> >
> > - **The primary challenge of iteration is finding a policy for effective exploration**, which is not trivial [3,4]. In this paper we use the rejection sampling and diversity selection to induce iterative exploration for effective and unique harmful instruction generation. Continuous iterative learning requires a careful balance between exploration and exploitation to prevent potential performance regression [5].
> >
> > >> **What if you generated 10000k instruction candidates and performed t=1 iteration versus 2000k instruction candidates with t=5 iterations?**
> >
> >
> >
> > When we have collected enough effective “failure” feedback with a diverse selection strategy, iterative optimization can significantly benefit the failure-inducing framework, as shown in Table 1. However, in knowledge-intensive tasks, e.g., math, the generated instructions from $t=1$ iteration is challenging enough for the target model. Future work can extend ReverseGen to explore effective training strategies for failure-priority data distributions.
> >
> >
> >
> > >> **Influence of the DPO training samples. What if you generate 1k or 10k samples to populate the positive and negative sets for DPO training?**
> >
> >
> >
> > ReverseGen remains robust and effective even with small datasets. For instance, when generating harmful instructions for Llama-2-7b-chat using OpenLLaMA-3B, we only collected 238 effective training samples from 50,000 prompts, which still yielded a 1.40x increase in ASR at t=1. With t=2, we collected 380 samples, leading to a 3.56x increase in ASR, and at t=3, with 598 samples, we achieved a 4.44x increase in ASR.
> >
> >
> >
> > In a less challenging scenario, when attacking Vicuna-7B-v1.5 with OpenLLaMA-3B, we collected 2,894 training samples from 50,000 prompts, resulting in a 2.95x increase in attack success rate (ASR) at $t=1$. We include these experimental results in Appendix E.2.
> >
> >
> > **Summary**
> >
> > - The failure-inducing reward is useful for data synthesis, especially for LLMs tuned for domain tasks. ReverseGen introduces a new perspective for scaling data.
> >
> > - When initial feedback data is limited, iterative optimization helps ReverseGen explore feedback based on self-generated instructions, gradually producing challenging data. During the iterations, exploring novel spaces is the key for continuous improvement.

---

> > > ### Author Response · Authors · 2024-11-27
> > >
> > > > **Q3: Did you observe any performance drops in your experiments when training with increasingly difficult samples?**
> > >
> > > Your concern here is extremely insightful! The target models are not trained from scratch or the base pre-trained models.  Instead, we consider a more challenging setting that all target models been pretrained or post-trained on the specific domains. Despite this, our method can still gradually explore samples target model tend to fail. This finding aligns with the data annealing strategy proposed in [1], i.e., annealing with upsamples high-quality and difficult data can boost the performance of pre-trained models.
> > >
> > > However, training models with 100% hard samples can destroy performance, e.g., for math reasoning, accuracy can drop from 79.89 to 76.46. To mitigate this issue, there are two strategies for training target model on hard data: (1) mixed general (or easy) samples with difficult samples and (2) biases more computation towards the difficult queries.
> > >
> > >  > **Q4: In Tables 1, 2, 3 you have a rows ‘without failure induction’ but you do not describe what this ablation is?**
> > >
> > > ReverseGen w/o failure induction refers to the proposer model fine-tuned on the initial instruction sets. We have clarified this notation in the main text for better understanding.
> > >
> > > [1] Llama Team, AI @ Meta. The Llama 3 Herd of Models. https://arxiv.org/abs/2407.21783
> > >
> > > [2] Yuxuan Tong, Xiwen Zhang, Rui Wang, Ruidong Wu, Junxian He. DART-Math: Difficulty-Aware Rejection Tuning for Mathematical Problem-Solving. https://arxiv.org/abs/2407.13690
> > >
> > > [3] Hanze Dong, Wei Xiong, Bo Pang, Haoxiang Wang, Han Zhao, Yingbo Zhou, Nan Jiang, Doyen Sahoo, Caiming Xiong, Tong Zhang. RLHF Workflow: From Reward Modeling to Online RLHF. https://arxiv.org/abs/2405.07863
> > >
> > > [4] Wei Xiong, Chengshuai Shi, Jiaming Shen, Aviv Rosenberg, Zhen Qin, Daniele Calandriello, Misha Khalman, Rishabh Joshi, Bilal Piot, Mohammad Saleh, Chi Jin, Tong Zhang, Tianqi Liu. Building Math Agents with Multi-Turn Iterative Preference Learning. https://arxiv.org/pdf/2409.02392
> > >
> > > [5] Llama 2: Open Foundation and Fine-Tuned Chat Models. https://arxiv.org/abs/2307.09288

---

> > > > ### Author Response · Authors · 2024-11-30
> > > >
> > > > Dear Reviewer u19j,
> > > >
> > > > We would appreciate it if you could let us know whether our response has adequately addressed your concerns and questions. We remain available to address any further questions you may have.
> > > >
> > > > Thank you once again for your time and effort!

---

> > > > > ### Author Response · Authors · 2024-12-02
> > > > >
> > > > > Dear Reviewer u19j,
> > > > >
> > > > > As the discussion period approaches its closing date, we wanted to check in to see if our responses have addressed your concerns. We’re happy to continue the conversation if needed.
> > > > >
> > > > > Thank you in advance for your support!

---

> > > > > > ### Author Response · Authors · 2024-12-03
> > > > > >
> > > > > > Dear Reviewer u19j,
> > > > > >
> > > > > > Thank you for your constructive comments, which are very helpful for improving our paper. We greatly value your feedback, but there are **less than 4 hours** remaining until the rebuttal deadline. We sincerely look forward to your new feedback.
> > > > > >
> > > > > > To summarize, in response to your suggestions, we have added:
> > > > > >
> > > > > > - An explanation of performance in the mathematical reasoning domain in Section 4.4 and Appendix G.1.
> > > > > > - A clarification of the role of iterative learning in the ReverseGen framework in Figure 2 and Appendix E.2.
> > > > > > - Additional ablation analyses concerning iteration times, DPO training size, and synthesis data size.
> > > > > > - Clarifications to improve the presentation and address any confusion.
> > > > > >
> > > > > > If our responses address your initial concerns, we would greatly appreciate it if you could consider updating your assessment.
> > > > > >
> > > > > > Best regards,
> > > > > >
> > > > > > The Authors of Paper 8852

---

### Author Response · Authors · 2024-11-27
**Updated Manuscript and Response to All Reviewers**

We sincerely thank all the reviewers for their feedback and constructive comments. We appreciate the reviewers' recognition of the novelty and significance of preference learning from failure feedback for data synthesis (R#1-u19j, R#2-29jz, R#3-HFrR, R#4-yaDx), the clear presentation of our paper (R#2), as well as the comprehensive and effective results across diverse tasks (R#1, R#2, R#4).

At the high level, we aim to:
1. investigate whether iteratively preference learning from model failures can effectively produce **new challenging samples**, particularly for models pre-trained on task-related data, and
2. evaluate the impact of training models on challenging samples by applying ReverseGen to three distinct tasks.  These guide our methodology and experimental design.

In response to the reviewers’ comments, we have conducted **four** additional experiments (including two baselines and two analyses) and revised the manuscript accordingly. **Changes are highlighted in blue in the updated PDF.** The updates are summarized as follows:

1. **Sections 2.1 & 3**: Added detailed explanations of prior works and revised the methods section for clarity, reducing implementation details (R#3).
2. **Section 4.1 & Appendix C**: Enhanced the reward process and parameters with a detailed paragraph and table (R#3, R#4).
3. **Figure 2**: Analyzed iteration times for ReverseGen performance (R#1, R#4).
4. **Table 3**: Clarified the experimental setup (R#3).
5. **Table 5**: Included results for the iterative LLM2LLM approach (R#1).
6. **Sections 4.4 & Appendix G.1**: Added explanations for math performance and provided a detailed results analysis (R#1, R#4).
7. **Appendix A**: Added motivation for failure-inducing exploration (R#4).
8. **Appendix E.2**: Provided analyses of iteration times and preference training data (R#1).
9. **Appendix G.2**: Included additional details on prompting powerful models (R#2).

We have addressed the queries and suggestions raised by each reviewer in detail and provided clarifications where necessary. We greatly appreciate the reviewers’ valuable contributions, which have helped us improve the quality of our manuscript. Should further clarification be required to assist in advancing our score, please do not hesitate to let us know.

Thank you once again for your review and feedback!

---

### Author Response · Authors · 2024-12-02

Dear Reviewers,

We sincerely appreciate the time and effort you have dedicated to evaluating our paper. Your insightful feedback and constructive comments have been invaluable in improving our manuscript and addressing areas of potential confusion. We have carefully addressed each of your comments, provided additional experiments, and updated the manuscript to clarify our points.

As the discussion deadline approaches (in less than 23 hours), we would be grateful to receive any additional feedback you may have.

Best regards,

The Authors of 8852

---

### Meta-Review · Area_Chair_TzXM · 2024-12-21

**Metareview:**

This paper presents ReverseGen, a novel approach for generating training data by identifying model weaknesses through failure-induced exploration. The method uses a proposer model fine-tuned via preference learning to generate challenging examples that expose target model failures. Key strengths include: (1) A novel framework for data synthesis through failure exploration, (2) Comprehensive evaluation across safety, honesty and math reasoning tasks, and (3) Demonstrated improvements over baselines. Main weaknesses initially centered on unclear theoretical justification, limited baseline comparisons, and analysis of computational costs. However, the authors provided extensive experimental results and analysis during rebuttal that adequately addressed these concerns. The paper makes a meaningful contribution to data synthesis for LLM training and warrants acceptance.

**Additional Comments On Reviewer Discussion:**

The discussion focused on three key aspects: (1) Theoretical motivation and framework clarity (u19j, yaDx), (2) Baseline comparisons and model selection rationale (29jz, HFrR), and (3) Analysis of iterative refinement and computational requirements (HFrR, yaDx). The authors thoroughly addressed these through additional experiments, ablation studies, and clarified explanations. Notably, they demonstrated ReverseGen's effectiveness with limited data (238 samples yielding 1.4x ASR improvement) and provided comprehensive comparisons with state-of-the-art baselines like LLM2LLM. All reviewers ultimately raised their scores to 6, acknowledging the paper's contributions while suggesting further improvements for the camera-ready version.

---

### Decision · Program_Chairs · 2025-01-22

Accept (Poster)